# Development and testing of a reduced carbohydrate intervention for the management of obesity and reduction of gestational diabetes (RECORD): protocol for a feasibility randomised controlled trial

Moscho Michalopoulou [1,2] Susan A Jebb,[1,2] Lucy H MacKillop [3,4] Pamela Dyson,[2,5] Jane E Hirst,[3,4] Amy Wire,[6] Nerys M Astbury [1,2]

For numbered affiliations see end of article.

**Correspondence to**
Moscho Michalopoulou;
moscho.michalopoulou@phc.ox.ac.uk

## ABSTRACT

**Introduction** Previous trials of dietary interventions to prevent gestational diabetes mellitus (GDM) have yielded only limited success. Low-carbohydrate diets have shown promise for the treatment of type 2 diabetes, but there is no evidence to support their use in pregnancy. The aim of this study is to explore the feasibility of a moderately reduced-carbohydrate dietary intervention delivered from mid-pregnancy alongside routine antenatal care.

**Methods and analysis** This is a feasibility randomised controlled trial (RCT) with embedded qualitative study. Sixty women who are pregnant <20 weeks' gestation, with body mass index ≥30 kg/m² at their antenatal booking appointment, will be randomised 2:1 intervention or control (usual care) and followed up until delivery. The intervention is a moderately reduced-carbohydrate diet (~130–150 g total carbohydrate/day), designed to be delivered alongside routine antenatal appointments. Primary outcomes are measures of adoption of the diet and retention of participants. Secondary outcomes include incidence of GDM, change in markers of glycaemic control, gestational weight gain, total carbohydrate and energy intake. Process outcomes will examine resources and management issues. Exploratory outcomes include further dietary changes, quality of life, maternal and neonatal outcomes, and qualitative measures.

**Ethics and dissemination** This trial was reviewed and approved by the South-Central Oxford B Research Ethics Committee NHS National Research Ethics Committee and the Health Research Authority (Reference: 20/SC/0442). The study results will inform whether to progress to a full-scale RCT to test the clinical effectiveness of the RECORD programme to prevent GDM in women at high risk. The findings will be published in peer-reviewed journals and presented at conferences.

**Trial registration number** ISRCTN16235884.

### STRENGTHS AND LIMITATIONS OF THIS STUDY

⇒ The intervention has been codeveloped by a multidisciplinary team of academic researchers and clinical professionals working in maternity, together with patients, to create a support programme likely to be acceptable to women and suitable for use in routine antenatal care.

⇒ Quantitative and qualitative analyses will assess the feasibility (adherence and retention rate) and suitability of reducing carbohydrate intake in mid-pregnancy, to inform any progression to a future full trial.

⇒ Dietary intake will be assessed at only two time points, and we will not be able to assess fluctuations in intake over time.

⇒ This study is not powered to detect clinical effectiveness of the dietary intervention, and subsequent full-scale trials will be required to assess whether this intervention can effectively attenuate gestational weight gain and improve glycaemic control early in pregnancy, with the prospect of preventing gestational diabetes.

## INTRODUCTION

Gestational diabetes mellitus (GDM) is defined as hyperglycaemia that develops or is first diagnosed during pregnancy.[1] GDM significantly increases the risk for adverse health outcomes for both the mother and the offspring,[2–9] In the UK, approximately 16% of women who are pregnant develop GDM, but the prevalence reaches up to 25%–30% in women living with obesity.[10] Excessive gestational weight gain (GWG) has also been proposed as a risk factor for GDM,[11] and several clinical trials in women with overweight or obesity during pregnancy, have tested whether interventions aimed at limiting GWG could help prevent GDM.[12–24] Although some trials have been successful

in attenuating GWG, the majority have failed to demonstrate a reduced risk of developing GDM.[25] [26]

The small number of trials that have shown effectiveness in GDM prevention so far included intensive face-to-face dietetic support and/or provision of key foods to study participants, with no indication of how these strategies could be delivered in routine practice.[14] [16] [18] [22] [23] For example, the Finnish Gestational Diabetes Prevention (RADIEL) study, reduced the incidence of GDM by 39% in women at high risk, but required extra face-to-face visits and a group session to deliver standard healthy eating and physical activity advice, which may not be manageable in routine practice given the amount of appointments women are already required to attend.[14] The ESTEEM and St Carlos trials implemented a Mediterranean-style diet supplemented with extra virgin olive oil and nuts, provided as part of the trial, from early or mid-pregnancy and showed a similar reduction in GDM risk.[16] [23] Nevertheless, wide scale implementation of a Mediterranean-style diet in the UK could be challenging due to limited understanding of the dietary pattern, differences in dietary habits and preferences, and the cost of extra virgin olive oil.[27–29]

The mechanism behind the success of some dietary interventions over others in preventing GDM remains unclear. Implementation at earlier stages of pregnancy seems to be key, but the optimum dietary composition is unknown. In the RADIEL study, there was only a small improvement reported in adherence to the diet recommendations in the intervention group compared with the control group.[14] In the Mediterranean dietary interventions, it was speculated that increased unsaturated fat and polyphenol consumption might have been beneficial with regards to insulin sensitivity and inflammation.[16] [23] However, a reduction in dietary carbohydrate as a natural result of higher fat and protein consumption, in conditions when energy intake remains similar, may have also played a role. Indeed, successful interventions to date included some advice for reducing sugar consumption and/or moderating starchy foods.

Low-carbohydrate diets are effective for the management of type 2 diabetes, especially over the short term (3–6 months).[30] Reducing carbohydrate intake from earlier stages of pregnancy may help reduce hyperglycaemia, independent of effects on weight gain, and may also help attenuate GWG by reducing energy intake. Although low-carbohydrate diets (<130 g/day)[31] are not recommended for women who are pregnant, due to safety concerns associated with ketosis, research in non-pregnant populations suggests that significant ketosis only occurs when carbohydrate intake is severely restricted to <50 g/day, leaving considerable scope to adopt a reduced-carbohydrate diet.[31] The level of carbohydrate reduction should be considered carefully though, particularly given that pregnancy is a ketosis-prone state, where ketones may develop at higher carbohydrate intakes than in non-pregnant populations.[32] One previous randomised controlled trial (RCT) specifically aimed at reducing total carbohydrate consumption as part of an intensive healthy eating intervention.[33] Despite a substantial significant effect on limiting GWG, the intervention did not reduce GDM incidence. This study was underpowered to detect difference in GDM risk, and it resulted in only a small reduction in carbohydrate intake compared with usual care (reduction of 4.8% of energy intake from carbohydrates, equivalent to about 30 g of carbohydrate per day). Thus, the effect of a reduced-carbohydrate diet as a preventative strategy for GDM remains to be investigated.

We have developed a dietary approach based on moderate carbohydrate reduction suitable for use in pregnancy, combined with behavioural support. The reduced-carbohydrate intervention for the management of obesity and reduction of gestational diabetes (RECORD) also recommends consumption of fresh foods, plenty of vegetables and moderate protein intake. It was designed with midwives and dietitians experienced in working with women who are pregnant, together with patients, and is designed to be delivered alongside routine antenatal appointments. The aim of this trial is to assess the feasibility of the intervention and whether it leads to a substantial reduction in total carbohydrate intake, defined as target mean intake of around 130–150 g/day, that is, reduction of around 80–100 g/day compared with the national average,[34] for the purposes of this study.

Other objectives include:
1. To assess the study procedures to inform the design of a full RCT.
2. To assess the potential effect of the reduced-carbohydrate behavioural intervention on markers of glycaemic control, incidence of GDM, GWG, dietary intake, quality of life, and maternal and neonatal outcomes.
3. To qualitatively analyse participant experiences of the intervention.

## Study design
A two-arm parallel group individually RCT, in women who are at <20 weeks' gestation with a body mass index (BMI) ≥30 kg/m². Participation will last approximately 6 months per participant; from early in the second trimester (no more than 19 weeks plus 6 days), until delivery. Participants will attend two study visits, one at baseline (<20 weeks' gestation) and one for follow-up, at 24–28 weeks' gestation.

## Recruitment
Participants will be recruited from one maternity care centre in Oxfordshire, UK. The study is expected to run from May 2021 to December 2022. Women attending the dating or nuchal ultrasound scan appointment (typically between 8 and 14 weeks' gestation) will be approached to take part in the study. Potentially eligible participants will be provided with the participant information sheet, and invited to give their contact details. A member of the research team will contact the women who were interested in taking part to discuss the study and assess self-reported eligibility to participate according to the full inclusion and exclusion criteria as detailed below.

Eligible individuals who wish to participate will be booked in for a face-to-face baseline appointment. The aim is to recruit women as early in pregnancy as possible, therefore, we will aim to book the baseline appointment for the trial within 2 weeks after the dating scan. However, we recognise this extra appointment at short notice may be challenging, and during this feasibility trial we will allow women to attend their baseline appointment up to 20 weeks' gestation. This allows at least a month to practise the dietary changes and assess whether the progression criteria are met. In addition, we will monitor the timing of the baseline appointment to check whether setting a limit at 14 weeks' gestation for recruitment to any future definitive trial, would be feasible. For those who decline to participate, permission will be sought to record the barriers to participation.

## Eligibility criteria
### Inclusion criteria

► Pregnant women <20 weeks' gestation.
► Aged ≥18 years.
► BMI ≥30 kg/m$^2$ measured at booking appointment.
► Planned antenatal care and delivery at the study setting.

### Exclusion criteria

The participant may be excluded if any of the following apply:

► Severe congenital fetal anomaly (confirmed by ultrasound).
► Planned termination of pregnancy or ectopic pregnancy.
► Prepregnancy diagnosis of diabetes or impaired glucose tolerance or taking metformin.
► Renal disease.
► Severe liver disease.
► Organ transplant.
► Cardiac failure (grade II New York Heart Association, and more severe).[35]
► Severe neurological disorder (including epilepsy).
► Severe psychiatric disease requiring in-patient admission.
► GDM diagnosis at the baseline study visit according to the National Institute for Health and Care Excellence (NICE) diagnostic criteria: a fasting plasma glucose level of 5.6 mmol/L or above, or a 2-hour plasma glucose level of 7.8 mmol/L and above.[36]
► History or presence of eating disorder.
► Hyperemesis gravidarum.
► Unable to understand spoken and written English.
► Previous bariatric surgery.
► Women participating in other intervention research studies.
► Any other significant disease or disorder which may either put the participants at risk because of participation in the trial, or may influence the result of the study, or the participant's ability to participate in the study.

## Study procedures

The baseline visit will take place before 20 weeks' gestation, at the hospital, with a member of the research team and lasts for approximately 2 hours. Written informed consent will be sought and eligibility for inclusion will be formally assessed. All participants will complete the following baseline assessments: blood pressure, weight and height measurements, a 2-hour 75 g oral glucose tolerance test (OGTT), a quality of life questionnaire and measurement of their carbohydrate intake as part of a broader assessment of their dietary intake via an online questionnaire. The participants will be randomised to one of the two trial arms. Participants in the intervention group will proceed to receive the intervention session. Due to the nature of the OGTT test and exclusion of women with GDM confirmed on this baseline test, full eligibility can only be confirmed after the visit, therefore after randomisation. While there is controversy over whether true GDM can be diagnosed prior to 24 weeks' gestation with existing criteria,[37] the clinical care team advised that women meeting the NICE criteria for GDM at the baseline visit,[36] should be considered to have GDM and be treated as such. All confirmed eligible participants will be invited to attend a further 2-hour research visit at 24–28 weeks' gestation where the same research measurements will be collected as a follow-up, and a second OGTT will be completed. Maternal and neonatal outcomes will be extracted from medical records after delivery. The study flow is in figure 1.

## Sample size

This feasibility study is not powered to detect a statistically significant difference in clinical effectiveness between the trial arms. Instead, we will test for reductions in carbohydrate intake as preliminary evidence of the adoption of the reduced carbohydrate diet to an extent that is expected to alter health outcomes. The average total carbohydrate intake in women who are not pregnant is about 230 g/day.[34] We have set the sample size to be able to detect a reduced target intake at 130–150 g, that is, ~140 g of total carbohydrate/day at follow-up, with 95% CI excluding 170 g/day. Progression criteria to a full trial also include adequate retention of participants as a measure of feasibility. We consider that a minimum of 80% of participants need to complete an OGTT at 24–28 weeks' gestation, similar to previous trials, with 95% CI excluding 70%.[12 16] Both progression criteria thresholds are satisfied with a sample size of 60 participants allocated in 2:1 ratio, intervention: control. Unequal randomisation allows us to have sufficient participants in the intervention arm for the qualitative study.

## Randomisation

Eligible participants will be randomised to one of the two treatment groups using permuted block randomisation with random block sizes of 3 and 6. An independent researcher will generate the allocation sequence and assignment will be revealed once eligibility has been confirmed using an online software (Study Randomizer)

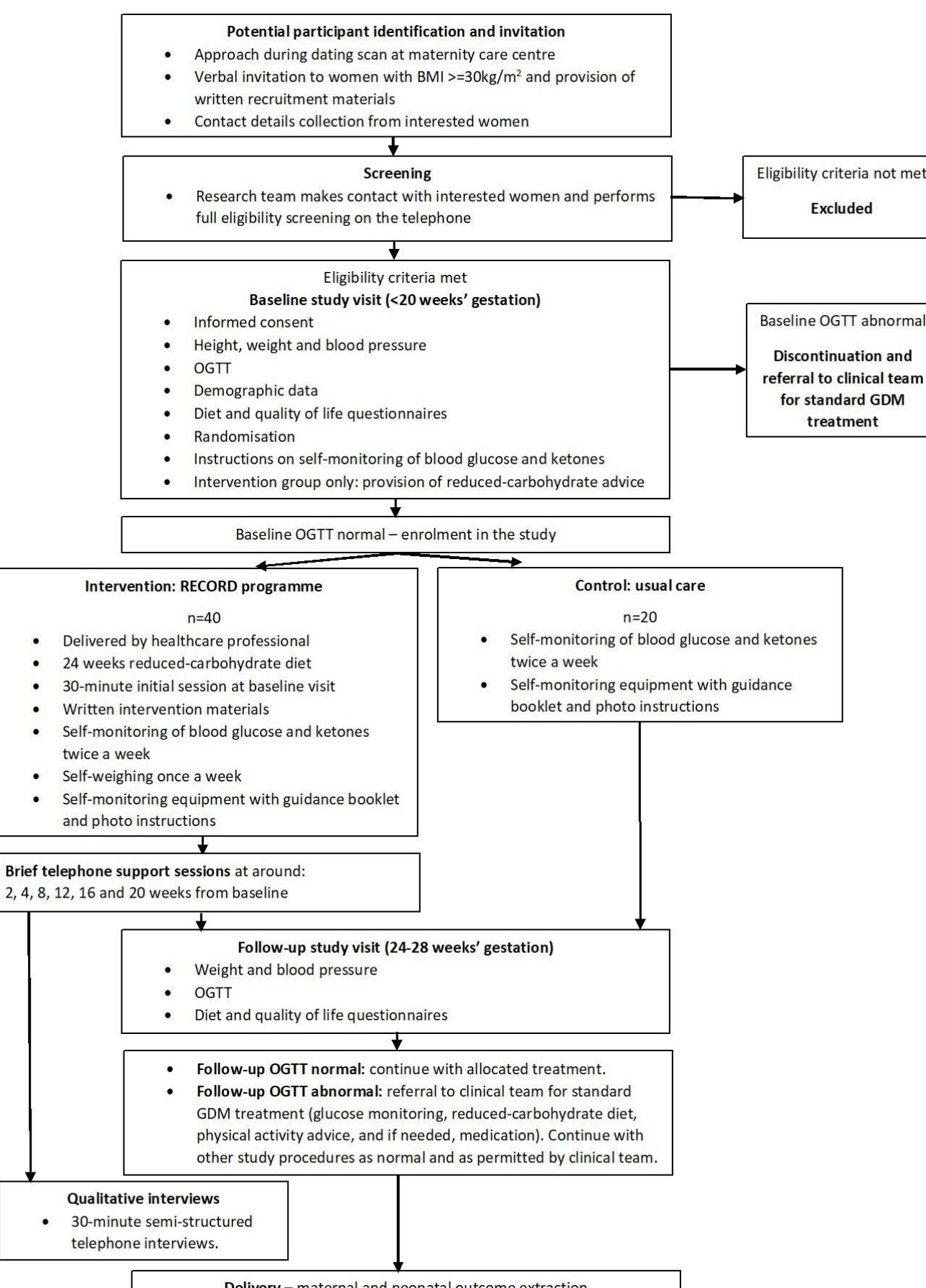

**Figure 1** Study flow. BMI, body mass index; GDM, gestational diabetes; OGTT, oral glucose tolerance test.

to ensure allocation concealment.[38] Due to the nature of this study, it will not be possible to blind the participants, clinicians or the study team to the treatment allocation beyond this point.

## Intervention and control

### Intervention: the record programme

The intervention comprises a moderately reduced-carbohydrate behavioural intervention, combining dietary advice, with structured written dietary information. It aims to improve participants' adherence to the programme through a simple behavioural support programme, which includes goal-setting, planning, feedback and problem-solving. The intervention will be delivered during a 30 min session with a healthcare professional (HCP) during the baseline visit, where the intervention materials and advice will be provided, and will be supplemented with up to six brief telephone support sessions of up to 15 min depending on patient need. Participants in this group will be advised to follow the intervention until delivery.

This intervention was developed based on a review of the literature, study protocols, qualitative data of similar interventions and interviews with a diabetes-specialist dietitian, two obstetricians, a research midwife and a patient representative with previous GDM. A group of women affected by GDM, also helped to shape the ideas for this research, and to determine the questions we are trying to answer, and the type of dietary information and support they felt would be helpful.

The dietary component of this intervention consists of a moderately reduced-carbohydrate diet (130–150 g of total carbohydrate per day), with no additional energy restriction. This is in line with the dietary advice provided in previous GDM trials, as well as for women diagnosed with GDM at the hospital. The difference here is that the reduced-carbohydrate advice is given earlier in pregnancy, to assess whether it has benefits for women at increased risk of GDM. The core principles include:

1. REFRAIN from having sugary foods and drinks (eg, biscuits, confectionery, juice, sugary fizzy drinks), with the exception of dairy and limited fruit intake.
2. REDUCE the amount of starchy carbohydrates to a moderate level, by having smaller portions of foods such as rice, pasta, bread, grains, potato, yams, cassava, plantain, at main meals.
3. REPLACE refined starchy carbohydrates with wholegrain, unrefined versions, and sugary foods and drinks, with lower-carbohydrate, and lower, or no-sugar alternatives.

We anticipate that women in this study will consume more than the target 130–150 g of carbohydrate per day at baseline, but if the diet questionnaire reveals that a participant has carbohydrate intake already close to the recommended level, the advice given will ensure that their intake is maintained through healthy food choices in line with the RECORD diet principles. Standard healthy eating advice during pregnancy is also included,

for example, high intake of vegetables, moderate intake of lean protein and guidance on foods that should be avoided during pregnancy. Information is discussed regarding the importance of controlling GWG, and women in this group are advised to weigh themselves once a week.

The development of this intervention was also informed by behavioural frameworks aiming to promote successful behaviour change. These are: the Capability, Opportunity and Motivation-Behavioural model, the Theoretical Domains Framework and the Behaviour Change Wheel.[39 40] By using these frameworks, we targeted the components of psychological capability (eg, knowledge/understanding, skills, decision making, self-regulation), physical and social opportunity (eg, social influences and support, and environmental changes) and reflective motivation (eg, knowledge/understanding, confidence regarding capabilities, beliefs regarding risks, intentions, plans). To address these components, the intervention includes:

► A consultation including the essential principles of the diet as described above, as well as the rationale behind the intervention and how it may work in controlling blood glucose levels, and attenuating GWG.
► Written resources to enhance participants' confidence and skills in following the programme, and to facilitate decision making, including advice on food selection, meal and snack suggestions, and suggested recipes.
► Setting personalised realistic goals, by letting participant choose what changes they are comfortable making and the steps to achieve them. Participants can choose to use a 'Personal action plan' booklet to set their goals and reflect on their progress, as well as goal progression checklists, and self-weighing charts, however, these are also addressed during the support sessions.
► Structured HCP follow-up to provide support, contingency planning, problem-solving, planning of social support, advice on lapses and help with environmental or emotional triggers, and to provide feedback on progress. Additionally, participants are able to self-monitor their health via home finger-prick testing of their blood glucose and ketones, and by weighing themselves, which provides additional monitoring.
► Provision of feedback on progress and changes in clinical measures as a result of following the intervention, for example, GWG, capillary blood glucose and ketones, glucose curve from OGTTs, blood pressure.

### Control

All women will receive routine antenatal care, which usually includes one-off face-to-face routine care NHS dietary advice during their booking appointment with a midwife. Anecdotal reports suggest this is focused on foods to be avoided during pregnancy for safety of the fetus, rather than healthy eating or weight control advice. This will have been given prior to randomisation in the

study. Participants allocated to the control group will receive no additional advice beyond this.

## Patient and public involvement

We convened two focus groups with eight women recruited from the local antenatal clinic, national GDM support website and a social media forum. The first group helped to prioritise and refine the research question. In a second phase, women were provided with an outline of the study proposal, and were asked to provide further comments. From both phases, it was clear that women felt upset, shocked and anxious following a diagnosis of GDM, and highlighted a need for preventative interventions that could be implemented earlier in pregnancy. Participants also highlighted the importance of including neonatal outcomes in addition to the maternal outcomes. Women were interested in taking part in clinical trials aiming to prevent, treat or manage GDM, however, felt that that attending extra appointments for study visits and in particular having to use the parking at the hospital, would be a major barrier to participation. As a final stage, a patient representative provided feedback on the intervention materials to help make them clearer and more concise and continues to support the study team.

## Outcomes
### Primary

The primary objective of this study is to test whether this reduced carbohydrate programme is feasible and leads to a reduction in carbohydrate intake in women who are pregnant and at risk of developing GDM. This feasibility measures will further determine whether to progress to a full RCT.

Progression criteria will be based on a traffic light system to decide whether to proceed to a full RCT (green), proceed with amendments for remediable issues (amber) or to not to proceed (red) (figure 2).[41] The progression criteria will be:

Adoption of the intervention: that the intervention group reduces their total carbohydrate consumption by the time of follow-up, with green light for 95% CI of average intake that does not cross 170 g/day. (Evaluated

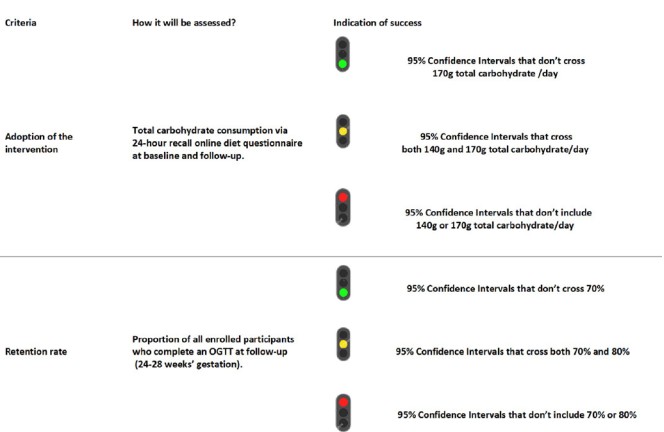

**Figure 2** Progression criteria. OGTT, oral glucose tolerance test.

as part of a broader assessment of dietary intake through an online 24-hour dietary recall questionnaire at the baseline and follow-up visits—detailed below).

Retention rate: proportion of all participants who complete an OGTT at the follow-up time point, with green light for 95% CI of average completion rate that does not cross 70%. (Evaluated by documentation at 24–28 weeks' gestation; assessed as the proportion of participants for whom OGTT measurements are available).

### Secondary

► Change in fasting plasma glucose, 1-hour and 2-hour glucose. (OGTT at baseline and follow-up visit).
► Change in glycated haemoglobin (HbA1c), fasting plasma insulin and insulin resistance (Homeostatic Model Assessment of Insulin Resistance-HOMA-IR). (Blood samples collected as part of OGTT at baseline and follow-up visit).
► Proportion of participants who are diagnosed with GDM. (Documentation at baseline and follow-up visit).
► GWG: weight change from baseline until before delivery. If predelivery weight is not available, we will use weight at 36 weeks' gestation collected as part of routine practice.
► Proportion of participants meeting the Institute of Medicine recommendations for GWG.[42]
► Dietary intake: total carbohydrate intake (g/day) and total energy intake (MJ/day), percentage of daily total energy intake from carbohydrates. (Dietary composition analysis at baseline and follow-up visit—see details below)
► Distribution of fasting capillary blood glucose and ketone levels across gestational age from baseline until before delivery.

### Process

► Number of eligible women who are approached and enrolled into the study per month. (Documentation)
► Percentage of intervention participants who complete the initial and support sessions. (Documentation at each relevant time point)
► Percentage of participants in each group who attend the follow-up visit. (Documentation)
► Percentage of participants for whom weight at 36 weeks' gestation is available (usually measured as part of a routine hospital appointment).
► Proportion of participants who record their blood glucose and blood ketones every week. (Documentation from baseline until before delivery)
► Proportion of participants in the intervention group who self-monitor their weight every week. (Documentation from baseline until before delivery)
► Proportion of participants in the control group who, after randomisation, follow a reduced-carbohydrate diet (contamination of the control group). (Dietary composition analysis at baseline and follow-up visit—see details below)

► Proportion of essential elements included in intervention delivery sessions (main and support sessions). (Evaluation of audiorecorded sessions against a checklist)

### Exploratory outcomes

► Further dietary changes: fibre intake, percentage of daily total energy intake from total fat, saturated fat and protein. (Dietary composition analysis at baseline and follow-up visit—see details below)
► Changes in systolic and diastolic blood pressure from baseline to follow-up.
► Changes in pregnancy-related outcomes for example, gestational hypertension and pre-eclampsia, requirement for insulin or oral diabetic medication, mode of delivery, induction of labour, gestational age at delivery, birth weight, sex of baby, rates of macrosomia (>4500 g),[43] large-for-gestational age (>90th percentile) and small-for-gestational age infant (<10th percentile),[43] neonatal adverse events, for example, neonatal intensive care unit admission, shoulder dystocia/birth trauma, neonatal hypoglycaemia requiring intravenous glucose, neonatal hyper bilirubinaemia requiring phototherapy. (Extraction from medical record following delivery)
► Change in quality of life from baseline to follow-up.
► Participant acceptability of the intervention and the materials, overall experience of the intervention, barriers and facilitators of adherence (qualitative interviews).

### Measurements

A schedule of measurements can be found in table 1.

### Sociodemographic characteristics

Participants will be asked to self-report highest education level, employment status and ethnicity. Age will be accessed from the medical records.

### Physical measurements

Height will be measured using stadiometers, to the nearest 0.1 cm. Weight will be measured to the nearest 0.1 kg using a digital scale (Active Era, BS-03S) at the study visits. For self-weighing at home, participants in the intervention group will be instructed to use their own digital scales or will be given a set by the research team if they do not own a set. They can either text the measurements to the research team or log them manually onto a mobile application (detailed below). At 36 weeks' gestation, women should be weighed during a routine antenatal appointment at their primary care practice, and digital scales are used for this. If predelivery weight is available, digital scales are also used for this. Blood pressure will be measured in duplicate using a digital blood pressure machine (Omron, M3) after 5 min seated rest, with at least 1 min between each measurements.

### OGTT and remote glucose and ketone monitoring

Women are asked to attend the study visits after an overnight fast, and to refrain from smoking, vaping, using nicotine replacement therapy or chewed sugary gum for at least 8 hours before. A fasting venous blood sample will be collected for the determination of glucose, HbA1c, insulin concentrations. Participants are provided with a 75 g glucose load (Polycal, Nutricia). Further venous blood samples will be collected at 1-hour and 2-hour postdrink, for the determination of glucose concentrations.

All participants are provided with a Bluetooth-enabled capillary blood glucose and ketone monitoring kit (4SURE Smart Duo, Nipro Diagnostics). The researcher provides a short demonstration and information flyer on how to collect and record the measurements, and advises participants to take a fasting capillary glucose and ketone measurement twice a week. The monitor is connected remotely to a mobile application (Diabetes: M, Sirma Medical Systems), allowing the research team to monitor the readings remotely and escalate concerns to the clinical team. We did not provide cut-offs to participants for fasting capillary glucose and ketone levels, due to lack of level references specific to pregnancy and before GDM diagnosis. Two obstetricians from the study team who are also members of the clinical care team interpreted the measurements on an individual basis and/or based on evidence from non-pregnant populations. The collection of this data during the feasibility study will guide thresholds that can be set for any future trial.

### Carbohydrate consumption

Participants will complete at baseline (<20 weeks' gestation) and follow-up (24–28 weeks' gestation) an online multiple-pass 24-hour recall, validated against interviewer-led multiple-pass recalls in adolescents and adults (Intake 24, Newcastle University).[44 45] This will be used to primarily assess carbohydrate consumption, but also broader dietary intake. By the time participants enrol in the study (mid-pregnancy), their dietary intake is likely to have stabilised compared with the fluctuations usually observed in the first trimester due to pregnancy symptoms. Intake 24 asks participants to report whether a reporting day was typical with regard to their dietary intake, and this will be taken into account in the analysis.

### Quality of life

Participants will be asked to fill in a questionnaire about their quality of life, via the European Quality of Life Five Dimension questionnaire and subscale.[46]

### Retention and withdrawal

This study has been designed to run alongside routine antenatal appointments with remote support elements, to minimise the burden associated with attending extra visits. The brief telephone support sessions with a dietitian will act as prompts for the participants to engage with the intervention and self-monitoring. Participants are asked to attend one visit in addition to routine antenatal

**Table 1** Schedule of study measurements

| | Study period | | | | | | | | |
|---|---|---|---|---|---|---|---|---|---|
| | Enrolment | Allocation | Study visits/sessions | | | | | | Delivery |
| **Time point** | Baseline=week 0 (<20 weeks' gestation) | Week 0 | Week 2 | Week 4 | ~Week 8 | Follow-up visit = ~Week 12 (24–28 weeks' gestation) | ~Week 16 | ~Week 20 | Week 24 |
| **Enrolment:** | | | | | | | | | |
| Eligibility screening | X | | | | | | | | |
| Informed consent | X | | | | | | | | |
| Randomisation | X | | | | | | | | |
| **Trial arms:** | | | | | | | | | |
| **RECORD programme** | X | X | ←——————————————— | | | X | | | X |
| + telephone support | | | X | X | X | X | X | X | |
| Usual care | X | X | ←——————————————— | | | X | | | X |
| **Assessments:** | | | | | | | | | |
| Demographics | X | | | | | | | | |
| Weight | X | | | | | X | | | |
| Height | X | | | | | | | | |
| Blood pressure | X | | | | | X | | | |
| OGTT | X | | | | | X | | | |
| EQ-5D | X | | | | | X | | | |
| Self-completed 24-hour dietary recall | X | | | | | X | | | |
| Fasting plasma insulin | X | | | | | X | | | |
| HbA1c | X | | | | | X | | | |
| Self-monitoring of fasting capillary glucose | | Twice a week for the study duration ←——————————————— | | | | | | | |
| Self-monitoring of fasting capillary ketones | | Twice a week for the study duration ←——————————————— | | | | | | | |
| Self-weighing (intervention group) | | Once a week for the study duration ←——————————————— | | | | | | | |
| SAEs | | ←——————————————— | | | | | | | X |
| Maternal and neonatal outcomes | | | | | | | | | X |

EQ-5D, EuroQol-5 Dimensions questionnaire; HbA1c, glycated haemoglobin; OGTT, oral glucose tolerance test; SAEs, serious adverse events.

care. All of the women who take part would be offered an OGTT at 24–28 weeks of pregnancy for the diagnosis of GDM. The 24–28 weeks follow-up study visit and OGTT, replaces the routine care OGTT and the results are shared with the clinical care team. To compensate for the time and inconvenience in taking part, we offer participants financial incentives for their participation.

Each participant will have the right to withdraw from the study at any time, or may be withdrawn at the discretion of the research or clinical care team. Participants will have the right to withdraw their collected data if they wish. Withdrawn eligible and enrolled participants will not be replaced in this feasibility study. We will only replace participants whose baseline OGTT is abnormal, as they will have consented and been randomised before OGTT results become available, but they will have essentially failed the full screening.

### GDM diagnosis

Participants who develop GDM or who are diagnosed with GDM at the 24–28 weeks' follow-up visit, will be referred to the clinical care team and offered the current standard medical treatment. In line with the NICE guidelines, this includes monitoring of blood glucose (two of the fasting measurements will continue to be used in the study), and diet and physical activity advice for the management of GDM.[36] More specifically, women with GDM are advised to manage carbohydrate intake, to limit their weight gain until delivery, and are encouraged to do 150 min of moderate intensity activity per week. Women in the intervention group, can choose to continue following the RECORD programme until delivery (at the discretion of the clinical care team). Women in the control group will receive advice to reduce their carbohydrate intake at this stage, and this will be noted. All participants with GDM, will be encouraged to continue monitoring their ketone levels twice a week for the remaining of pregnancy.

### Statistical analysis

Descriptive statistics will be used to report the baseline characteristics of each trial arm. Continuous variables will be reported using mean/median and SD/range, while categorical variables will be reported using frequencies and percentages.

Since this is a feasibility study, the primary outcomes are the main progression criteria which will inform the design of a future RCT, powered to detect a significant difference in the efficacy of the intervention in preventing GDM. Descriptive statistics with 95% CIs will be used to analyse and report the progression criteria. For the analysis of the secondary, process-related and exploratory measures, all data collected up to each time point will be used. Descriptive comparative statistics will be used to analyse and report these measures along with 95% CIs (eg, difference in means or proportions between baseline and follow-up for each group or between the two groups). Data from audiorecorded and transcribed qualitative interviews on acceptability and experience of the intervention by participants will be analysed using thematic analysis and reported descriptively.

### Qualitative study

A qualitative substudy with participants randomised to the RECORD group will explore participants' acceptability of the intervention and their overall experience of the study, explore barriers and facilitators to adherence, and identify suggestions that could refine the intervention should it progress to a full RCT.

During the follow-up study visit, a member of the research team will provide participants allocated to the intervention group, with a separate participant information sheet about the qualitative interviews. If participants are happy to have this in-depth discussion about their experiences of the RECORD intervention, a member of the research team will arrange a convenient day and time for a telephone interview. The participants will consent verbally on the phone and the researcher will fill in written consent form. The interviews will last approximately 30 min and all interviews will be audiorecorded and transcribed verbatim. Data will be analysed using thematic analysis. We will conduct interviews with up to 30 participants, or until data saturation is reached.

### Trial management and monitoring

The day-to-day management of the study will be coordinated by MM. As this is a small feasibility study, with no adverse event monitoring or stopping rules, there are no monitoring committees in place. A trial management group consisting of the authors of this paper, a research midwife and a patient representative will have oversight of the trial and meet bimonthly to discuss progress and decide on actions to be taken.

### Adverse events

We will record and report all serious adverse events (SAEs) according to Good Clinical Practice (GCP) and Health Research Authority's (HRA) processes. Recording of SAEs will start immediately following enrolment into the study and will be continued until the end of the study.

### Data management

Data will be kept in accordance with GCP, the Data Protection Act 2018 and the General Data Protection Regulation. Two separate databases will be created, one containing all participant identifiable information and one where data collected during the research procedures will be entered in an anonymised manner, using a unique participant ID (electronic case report forms). Anonymised data from the online diet questionnaire will be downloaded from the Intake24 website and added to the second database. Quality of life anonymised data will be collected on hard copy questionnaires and then entered into the second database. The two databases, as well as anonymised recordings and transcriptions from the main and support intervention sessions, will be password protected and stored on the secure drive of Nuffield Department of Primary Care Health Sciences, and will

only be accessible to authorised members of the research team. After publication of the results has been sent out to participants, their contact details will be deleted/destroyed. We will retain the anonymised research data for future secondary analyses.

Direct access to study data will be granted to authorised representatives from the sponsor and host institution for monitoring and/or audit of the study to ensure compliance with regulations. Otherwise, confidentiality will be maintained and no-one outside the research team will have access to the database.

## ETHICS AND DISSEMINATION

This protocol (Version 1.3, 12 July 2021) was reviewed and approved by the South-Central Oxford B Research Ethics Committee NHS National Research Ethics Committee and the HRA (Reference: 20/SC/0442), and prospectively registered on ISRCTN (ISRCTN16235884). Any substantial changes to the protocol will be notified and reviewed by NHS Research Ethics Committee, the HRA and the Sponsor (University of Oxford) and trial registry will be updated accordingly. The findings of this feasibility study will be submitted for publication in a peer-reviewed journal, and presented at conferences, to disseminate the results to academic and health professional audiences, and made available to participants and to the wider public on our website at the time of publication.

**Author affiliations**
¹Nuffield Department of Primary Care Health Sciences, University of Oxford, Oxford, UK
²NIHR Oxford Biomedical Research Centre, Oxford, UK
³Nuffield Department of Women's and Reproductive Health, University of Oxford, Oxford, UK
⁴Oxford University Hospitals NHS Foundation Trust, Oxford, UK
⁵Oxford Centre for Diabetes, Endocrinology and Metabolism (OCDEM), University of Oxford, Oxford, UK
⁶Berkshire Healthcare NHS Foundation Trust, Bracknell, UK

**Acknowledgements** We thank Sr Yvonne Kenworthy, Research Midwife, for providing feedback on the intervention materials.

**Contributors** MM, NMA and SAJ developed the concept for the study and wrote the first draft of the protocol with input from LHM and JEH on clinical aspects. MM, NMA and SAJ prepared the study documents and coordinated the HRA and ethics application. The sponsor has reviewed all participant-facing documents as part of the ethics application. LHM, JEH and PD approved the final protocol. MM, NMA, SAJ and PD were involved in the detailed design of the intervention. MM and PD developed the intervention materials with input from NMA, SAJ, LHM, JEH and AW. MM drafted the manuscript for publication, with input from NMA and SAJ. LHM, JEH, PD and AW reviewed, provided input and approved the final manuscript.

**Funding** This trial is funded by the NIHR Oxford Biomedical Research Centre (IS-BRC-1215-20008). MM's time on this project is funded by Oxford-Medical Research Council Doctoral Training Partnership (MR/N013468/1). JEH is supported by a UK Research and Innovation Future Leaders Fellowship (MR/T040750/1). Sponsor: This trial is sponsored by the University of Oxford, Clinical Trials and Research Governance, Joint Research Office, Block 60, Churchill Hospital, Old Road, Headington, Oxford, OX3 7LE, UK.

**Competing interests** LM is a part-time employee of EMIS Group plc.

**Patient and public involvement** Patients and/or the public were involved in the design, or conduct, or reporting, or dissemination plans of this research. Refer to the Methods section for further details.

**Patient consent for publication** Not applicable.

**Provenance and peer review** Not commissioned; externally peer reviewed.

**ORCID iDs**
Moscho Michalopoulou http://orcid.org/0000-0002-6063-3307
Lucy H MacKillop http://orcid.org/0000-0002-1927-1594
Nerys M Astbury http://orcid.org/0000-0001-9301-7458

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
