## [Reviewer comments · BMJ Open]

ARTICLE DETAILS

TITLE (PROVISIONAL)	Development and testing of a reduced carbohydrate intervention for the management of obesity and reduction of gestational diabetes (RECORD): protocol for a feasibility randomised controlled trial
AUTHORS	Michalopoulou, Moscho; Jebb, Susan; MacKillop, Lucy; Dyson, Pamela; Hirst, Jane E; Wire, Amy; Astbury, Nerys

VERSION 1 – REVIEW

REVIEWER	J Puder Universite de Lausanne, DFME
REVIEW RETURNED	23-Feb-2022

GENERAL COMMENTS	The review was completed together with Sybille Schenk, dietitian at the Centre Hospitalier Universitaire Vaudois (CHUV), Lausanne, Switzerland. Reduced carbohydrate intervention for the management of obesity and reduction of gestational diabetes (RECORD): protocol for a feasibility randomised controlled trial This methods paper explores the feasibility of a reduced carbohydrate intervention in pregnancy in women with pre-existing obesity. The paper is well written and the subject relevant and most of the methodology precise. Overall, this is a nice and important study. However, there are some major and minor comments. MAJOR COMMENTS Title: The title is misleading and does not contain the primary outcome (something in the direction of feasibility; incidence of GDM and weight gain are not even secondary outcomes, but only exploratory outcomes) and should be adapted by the authors. At the moment that the protocol is submitted for review, the recruitment is halfway finished. This is a limitation for a RCT, specifically regarding the precise definition of primary and secondary outcomes. The authors should add this in the limitations. In addition, everything is written in a “future mode”, but some of it concerns the past. Strengths and limitations: Line 45-59: Similarly, the strengths and limitations do not reflect the primary and secondary outcomes and are too much focused on the exploratory outcomes; the authors should reword. Introduction: Line 67/68: We would ask the authors to add references. Line 69 to 78: The authors explained why other studies like ESTEEM could be difficult to implement in routine practice but do
--

not discuss why the RADIEL is not manageable in routine practice. For instance, it would be interesting to state that maybe patients had too many appointments. This could be later a reason why RECORD goes for phone calls.

Line 95: The authors should mention that pregnancy is also a ketoses-prone state and thus ketones may develop at higher carbs intake than in non-pregnant populations.

Line 109: The authors could precise the primary aim; what would be a significant reduction in carbohydrates intake?

Methods:

Line 149 & 168: We would ask the authors how can GDM be diagnosed <20 weeks of GA? Definitions before 24 weeks of GA are still a matter of controversy.

Line 165-166 & 280: Regarding the dietary intake, we would ask the authors to give more details about the assessment. For example, what if the women already have a carbohydrate intake around 130-150g? Thus, knowing that there are many changes and fluctuations of food intake in pregnancy, how is the Intake24 supposed to take into consideration the dietary changes that can occur in the first trimester before the intervention? Furthermore, what if the specific days (before intervention and the second time) when the women are filling up the Intake24 they eat more or less carbohydrates than usual, how is the tool measuring variations? Moreover, the 95CI takes up to 170g of carbohydrates but how is this really assessed by the team between the visit 1 and visit 2? Is it just calculated based on the 24-hour recalls or is there other information? Is the person calling the women in between the visits filling something up to assess the changes as well?

Line 179 & 280: We would ask the authors how do they exactly define primary outcomes as measures of adaption; what would be the mean carbs content to have success (130/140/150g)? Or what percentage of patients should attain the 130-150g of carbs and what would be the exact cut-off (130/140/150g)? Thus, when the authors describe a CI up to 170g of carbs, is this amount considered a modified low-carb (being close to the minimum of 175g recommended)? What would be the defined retention of less than 130/140/150g?

Line 212: The authors mention rightly in the introduction that an earlier start to intervene is the key and that recruitment is at the US appointment (8-14 weeks). Would they also recruit earlier and why include beyond 14 weeks (first visit/recruitment being before 20 weeks)? We would ask the authors to discuss this matter.

Line 254: We would like the authors to describe in details the standard healthy diet advice for the control group. How is this more or less standardized?

Line 311: We would like the authors to explain why not the total GWG?

Line 438: The authors should precise if they will also adapt the registry accordingly.

MINOR COMMENTS

Line 92: We would find it interesting if the authors would compare with the minimum of 175g of carbohydrates stated in the ADA guidelines.

Line 117: We would find it interesting to know the subcategories of obesity, in order to know if the feasibility and adherence of the women are different between women grade 1, 2 or 3 of obesity.

	Line 192: The authors say “some of the study team”. We would like them to be more precise about who exactly will be blinded. Does that mean that some people can be blinded or will be blinded? Line 215-224: It seems to be what many centers do as standard care for the early GDM. We would ask the authors to explain what would be novel? Quansah DY, Gross J, Gilbert L, Pauchet A, Horsch A, Benhalima K, Cosson E, Puder JJ. Cardiometabolic and Mental Health in Women With Early Gestational Diabetes Mellitus: A Prospective Cohort Study. J Clin Endocrinol Metab. 2022 Feb 17;107(3):e996-e1008. doi: 10.1210/clinem/dgab791. PMID: 34718650. Line 249-251: We would ask the authors to add details concerning the ketones levels references used. It is available for the glucose measurements but not the ketones. If not available, we recommend that the authors discuss the difficulties to find one related to pregnancy. Line 347: Very nice tool!
--	--

VERSION 1 – AUTHOR RESPONSE

The review was completed together with Sybille Schenk, dietitian at the Centre Hospitalier Universitaire Vaudois (CHUV), Lausanne, Switzerland.

Reduced carbohydrate intervention for the management of obesity and reduction of gestational diabetes (RECORD): protocol for a feasibility randomised controlled trial

This methods paper explores the feasibility of a reduced carbohydrate intervention in pregnancy in women with pre-existing obesity.

The paper is well written and the subject relevant and most of the methodology precise. Overall, this is a nice and important study. However, there are some major and minor comments.

MAJOR COMMENTS

Title: The title is misleading and does not contain the primary outcome (something in the direction of feasibility; incidence of GDM and weight gain are not even secondary outcomes, but only exploratory outcomes) and should be adapted by the authors.

At the moment that the protocol is submitted for review, the recruitment is halfway finished. This is a limitation for a RCT, specifically regarding the precise definition of primary and secondary outcomes. The authors should add this in the limitations. In addition, everything is written in a “future mode”, but some of it concerns the past.

Strengths and limitations:

Line 45-59: Similarly, the strengths and limitations do not reflect the primary and secondary outcomes and are too much focused on the exploratory outcomes; the authors should reword.

Introduction:

Line 67/68: We would ask the authors to add references.

Line 69 to 78: The authors explained why other studies like ESTEEM could be difficult to implement in routine practice but do not discuss why the RADIEL is not manageable in routine practice. For instance, it would be interesting to state that maybe patients had too many appointments. This could be later a reason why RECORD goes for phone calls.

Line 95: The authors should mention that pregnancy is also a ketoses-prone state and thus ketones may develop at higher carbs intake than in non-pregnant populations.

Line 109: The authors could precise the primary aim; what would be a significant reduction in carbohydrates intake?

Methods:

Line 149 & 168: We would ask the authors how can GDM be diagnosed <20 weeks of GA? Definitions before 24 weeks of GA are still a matter of controversy.

Line 165-166 & 280: Regarding the dietary intake, we would ask the authors to give more details about the assessment. For example, what if the women already have a carbohydrate intake around 130-150g? Thus, knowing that there are many changes and fluctuations of food intake in pregnancy, how is the Intake24 supposed to take into consideration the dietary changes that can occur in the first trimester before the intervention? Furthermore, what if the specific days (before intervention and the second time) when the women are filling up the Intake24 they eat more or less carbohydrates than usual, how is the tool measuring variations? Moreover, the 95CI takes up to 170g of carbohydrates but how is this really assessed by the team between the visit 1 and visit 2? Is it just calculated based on the 24-hour recalls or is there other information? Is the person calling the women in between the visits filling something up to assess the changes as well?

Line 179 & 280: We would ask the authors how do they exactly define primary outcomes as measures of adaption; what would be the mean carbs content to have success (130/140/150g)? Or what percentage of patients should attain the 130-150g of carbs and what would be the exact cut-off (130/140/150g)? Thus, when the authors describe a CI up to 170g of carbs, is this amount considered a modified low-carb (being close to the minimum of 175g recommended)? What would be the defined retention of less than 130/140/150g?

Line 212: The authors mention rightly in the introduction that an earlier start to intervene is the key and that recruitment is at the US appointment (8-14 weeks). Would they also recruit earlier and why include beyond 14 weeks (first visit/recruitment being before 20 weeks)? We would ask the authors to discuss this matter.

Line 254: We would like the authors to describe in details the standard healthy diet advice for the control group. How is this more or less standardized?

Line 311: We would like the authors to explain why not the total GWG?

Line 438: The authors should precise if they will also adapt the registry accordingly.

MINOR COMMENTS

Line 92: We would find it interesting if the authors would compare with the minimum of 175g of carbohydrates stated in the ADA guidelines.

Line 117: We would find it interesting to know the subcategories of obesity, in order to know if the feasibility and adherence of the women are different between women grade 1, 2 or 3 of obesity.

Line 192: The authors say "some of the study team". We would like them to be more precise about who exactly will be blinded. Does that mean that some people can be blinded or will be blinded?

Line 215-224: It seems to be what many centers do as standard care for the early GDM. We would ask the authors to explain what would be novel? Quansah DY, Gross J, Gilbert L, Pauchet A, Horsch A, Benhalima K, Cosson E, Puder JJ. Cardiometabolic and Mental Health in Women With Early Gestational Diabetes Mellitus: A Prospective Cohort Study. J Clin Endocrinol Metab. 2022 Feb 17;107(3):e996-e1008. doi: 10.1210/clinem/dgab791. PMID: 34718650.

Line 249-251: We would ask the authors to add details concerning the ketones levels references used. It is available for the glucose measurements but not the ketones. If not available, we recommend that the authors discuss the difficulties to find one related to pregnancy.

Line 347: Very nice tool!

VERSION 2 – REVIEW

REVIEWER	J Puder Universite de Lausanne, DFME
REVIEW RETURNED	02-Aug-2022
GENERAL COMMENTS	We thank the authors that responded to all of the comments and

	also accepted many of the proposals. We look forward to seeing the results !
--	---